# Study on Reciprocating Loading Tests and Moment-Rotation Theory of Straight-Tenon Joints in Traditional Wooden Structures

**Shibin Yu [1,2]**, **Wen Pan [1,2,*]**, **Hexian Su [1,2]** and **Liaoyuan Ye [2,3]**

1 Faculty of Civil Engineering and Mechanics, Kunming University of Science and Technology, Kunming 650500, China; 20191110010@stu.kust.edu.cn (S.Y.); shx870@kust.edu.cn (H.S.)
2 Yunnan Seismic Engineering Technology Research Center, Kunming University of Science and Technology, Kunming 650500, China; yly@163.com
3 Communist Party Committee Office, Yunnan Normal University, Kunming 650500, China
* Correspondence: 13312039@kust.edu.cn

**Abstract:** For the study of the mechanical properties of straight-tenon joints in traditional wooden structures, three specimens of T-shaped straight-tenon joints were made according to actual structures and subjected to reciprocating loading tests. The variation rules of different seismic performance indexes such as moment-rotation hysteresis curve, skeleton curve, stiffness, and energy dissipation capacity of the specimens were analyzed through tests. Based on the geometric deformation and static equilibrium conditions, the moment-rotation theoretical model of straight-tenon joints is derived and compared with the experimental results. The studies show that the hysteresis curve of joints under reciprocating loading consists of four stages: ascending, stress relaxation, descending, and sliding. The moment capacity of joints increases gradually with the rotational deformation, but the internal gap of the joints increases synchronously, resulting in a serious attenuation of the stiffness. Tenon and mortise plastic extrusion deformation and friction can dissipate energy, as the rotational deformation increases energy consumption, while the hysteresis loop "pinch" effect is more serious, and the equivalent viscous damping coefficient is gradually reduced. The prediction results of the joint moment-rotation theoretical model are closer to the experimental results, which can provide a theoretical basis for the overall seismic analysis of traditional wooden structures.

**Keywords:** mortise and tenon joint; reciprocating loading tests; moment-rotation model; hysteresis curve

## 1. Introduction

The construction method of traditional wooden structures is to cut and process the two beam ends into tenons, make mortises at specific positions on the column, and mechanically assemble the components with tenon and mortise to form the overall frame, as shown in Figure 1. The mortise and tenon joint plays an important role in traditional timber structures; not only can they transmit the gravity load on the beam but also exhibit certain bending resistance and energy dissipation under cyclic loading [1–3]. The failure of the mortise and tenon joint, a vital part of the structure, will lead to the overall tilt or even collapse of the structure, which seriously threatens the safety of occupants [4,5]. Therefore, studying the force mechanisms of mortise-tenon joints is a prerequisite for the overall anti-seismic analysis of traditional wooden structure systems and a theoretical basis for the structural fortification and renovation of such buildings.

In order to better preserve the valuable cultural heritage of traditional wooden buildings, many scholars have conducted extensive research on the mechanical properties of mortise-tenon joints in recent decades.

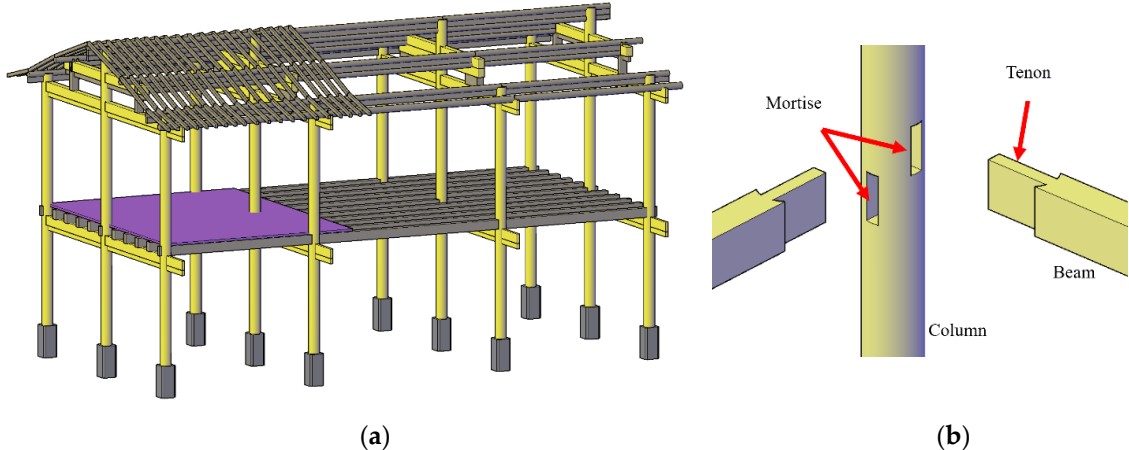

**Figure 1.** Traditional timber-framed structure: (**a**) Overall model; (**b**) The detailed structure of joints.

Experimental studies include Pang et al. [6], which conducted tests on the effect of mortise and tenon shoulders on the mechanical properties of dovetail joints and concluded that the mortise and tenon shoulders had significant effects on the failure mode, moment bearing capacity, and joint rigidity of dovetail joints under static loading. Gao et al. [7] quantitatively analyzed the effect of friction on the energy dissipation capacity of joints by conducting cyclic loading tests on through-tenons, dovetail tensions, and through-tenon with square-column below the beam. The results show that the coefficient of friction has an effect on both the energy dissipation capacity and the stiffness of mortise and tenon joints. Chen [8–12] conducted monotonic loading tests on scale joint models, such as straight-tenons, through-tenons, short column pin tenons, and dovetail tenons, to study the bending resistance properties of different mortise and tenon joints. Chun et al. [13–16] conducted low-cycle reciprocating loading tests on steamed bun-shaped tenons, through tenons and half tenons in traditional wooden structures in southern China, and analyzed the failure mode, hysteresis curve, skeleton curve, and angel rigidity of various types of mortise-tenon joint.

Theoretical model studies include Jiang et al. [17], which proposed a method to evaluate the rigidity of traditional mortise-tenon joints based on statistical process control charts. Xue [18], Zhang [19], Ma [20], and He et al. [21] took the through-tenon joints in traditional Chinese wooden structures as the research object and successively proposed different moment-angel models by analyzing their force mechanisms according to the joint structure characteristics. Xie et al. [22,23] derived the moment-rotation theoretical equations for one-way straight-tenon and dovetail joints and analyzed the effects of geometry and the friction coefficient on moment bearing capacity. Pan et al. [24,25] analyzed the force characteristics of straight-tenon joints in traditional wooden structures under low-cycle reciprocating load. They established the $M$-$\theta$ Mechanical model, considering the removed tenons' influence by analyzing the mechanical characteristics of straight-tenon joints in traditional wooden structures under low cycle reciprocating loads. Then, with the elastic, yield, and load limit points as characteristic points, a tri-linear multi-parameter moment-rotation mechanical model was proposed by taking dovetail tenon and through tenon as the research objects.

Although there are many experimental and theoretical studies on mortise-tenon joints, the theoretical aspects still need to be further developed. In the literature [18–20,22,23], the theoretical models of mortise and tenon joints are related not only to the parameters of joint geometry and the mechanical properties of the material, but also to the position of the external loads. These models are only applicable to the case of joint deformation caused by concentrated forces, while other conditions have great limitations. In addition, the center of rotation of mortise and tenon nodes is based on the intersection of beam and column axes as the center of rotation in the literature [21,24,25], which is not consistent with

actual deformation. Therefore, this paper discusses the low-cycle reciprocating loading test on mortise-tenon joint specimens. It analyzes the variations in joint deformation, hysteresis curve, skeleton curve, stiffness, and energy dissipation capacity of the test specimens. According to the rules of mortise and tenon node deformation in the test, a model of bending moment-rotation relationship is established through static equilibrium conditions. The model contains parameters such as geometry, transverse elastic modulus, and compressive strength, which can be used to predict the bending moment bearing capacity of the node when the rotation angle is $\theta$. It is more versatile than the existing methods. Through comparison, it is found that the predictions of the model closely match the experimental results, providing a theoretical basis for the design of traditional timber structures.

## 2. Materials and Methods

### 2.1. Wood Properties Testing

The straight-tenon joint specimens were made of Yunnan pine, whose average density and moisture content were 0.493 g/cm$^3$ and 17.2% after the test measurement. The mechanical properties tests were carried out in strict accordance with the relevant Chinese national standards, and the test photos are shown in Figure 2.

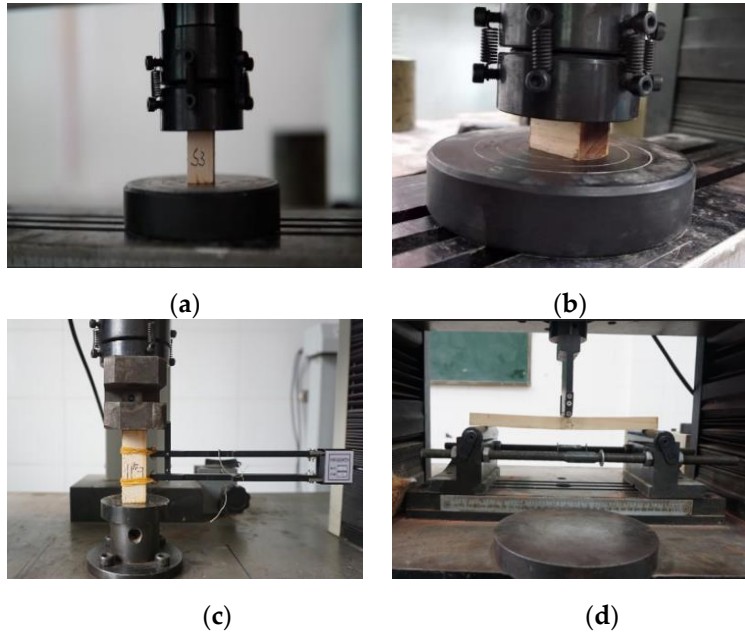

**Figure 2.** The test of wood mechanical properties. (**a**) Parallel-to-grain compressive strength test; (**b**) Perpendicular-to-grain compressive strength; (**c**) Elasticity modulus test; (**d**) Bending strength test.

Referring to the requirements of Chinese standards GB/T 15777-2017 [26] and GB/T 1927.11-2022 [27], compressive modulus of elasticity and compressive strength of wood in the direction of parallel-to-grain were tested. The size of the specimen was 20 mm × 20 mm × 30 mm, and 30 mm was the direction of parallel-to-grain. Compressive modulus and compressive strength testing of wood in the direction of the transverse grain, in accordance with the requirements of GB/T 1927.13-2022 [28] and GB/T 1927.12-2021 [29] was conducted, and the size of the specimen was 20 mm × 20 mm × 30 mm, and 30 mm is the direction of the perpendicular-to-grain. GB/T 1927.9-2021 [30] stipulates that the size of flexural strength specimen is 300 mm × 20 mm × 20 mm, and 300 mm is the direction of parallel-to-grain. The loading point is located in the middle of the specimen, the distance between the two ends of the support point was 270 mm, as shown in Figure 2d.

Nine samples were used for each mechanical property test and the results are shown in Figure 3. According to the corresponding specification, the compressive strength of the

parallel to grain is the maximum value in the test curve, while the compressive strength of the perpendicular to grain is the proportional limit point of the test curve, and the yield point is determined by the method in the standard [29], as shown in Figure 4. The slope of the stress-strain curve corresponding to loading from 10%–40% of the ultimate load is the modulus of elasticity. The mechanical properties of wood are the average of the test results of the specimens, as shown in Table 1.

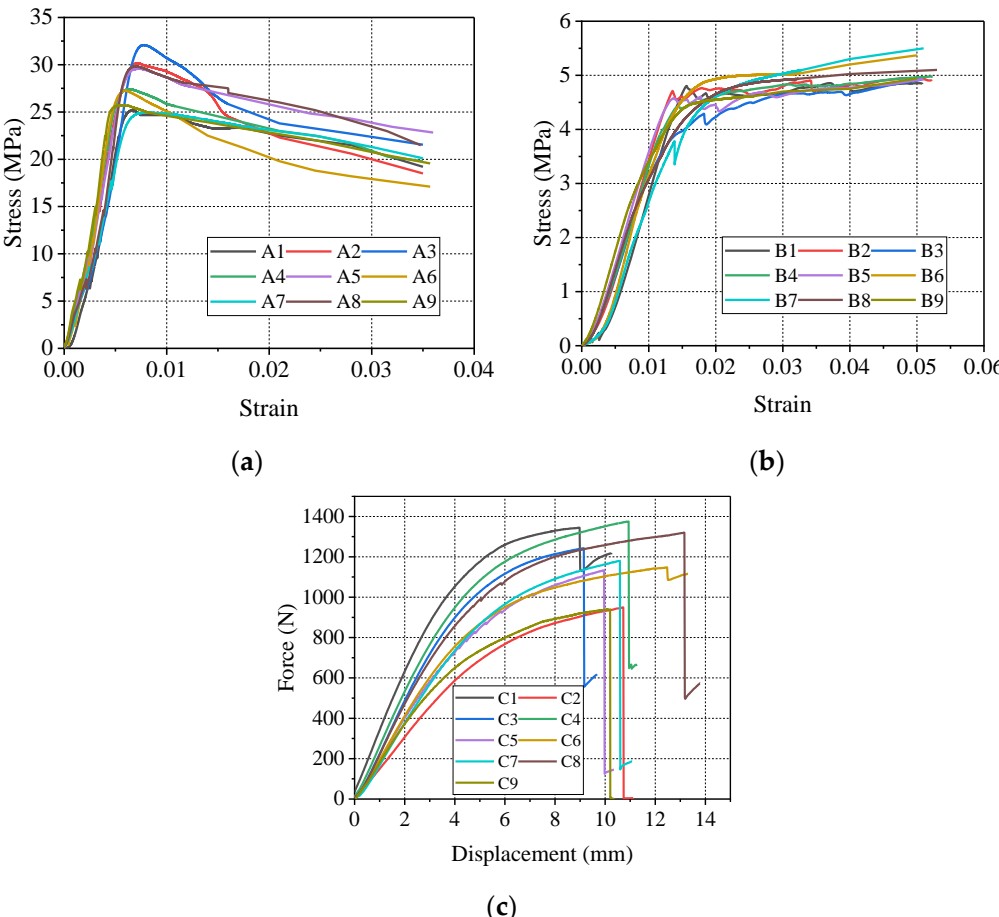

**Figure 3.** Physical and mechanical properties of wood materials. (**a**) Stress-strain curve of parallel-to-grain compression; (**b**) Stress-strain curve of perpendicular-to-grain compression; (**c**) Load-displacement curves in wood bending strength tests.

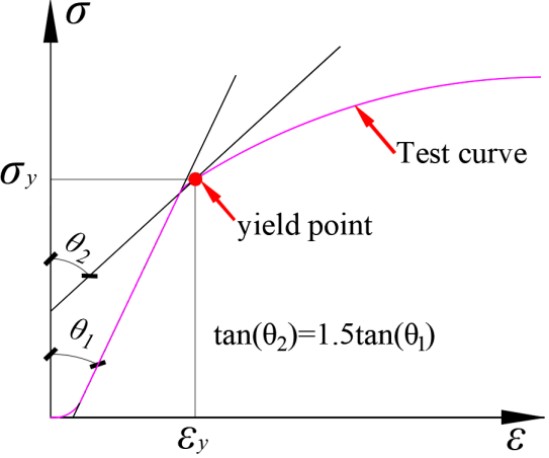

**Figure 4.** Methods used for the estimation of the yield point.

**Table 1.** Parameters for mechanical properties of wood.

| $E_s$/MPa | E/MPa | $f_s$/MPa | $f_y$/MPa | $f_m$/MPa |
|---|---|---|---|---|
| 10,732.4 | 375.8 | 26.3 | 4.6 | 59.8 |

Note: $E_s$ is the parallel-to-grain modulus of elasticity, E is the perpendicular-to-grain modulus of elasticity, $f_s$ is the parallel-to-grain compressive strength, $f_y$ is the perpendicular-to-grain compressive strength, $f_m$ is the bending strength of wood.

## 2.2. Mortise–Tenon Joint Specimens

Based on the research data of Yikeyin-styled (seal-shaped) traditional wooden houses in Tonghai County, Yunnan Province, three identical T-shaped straight mortise-tenon joints, numbered ZS1, ZS2 and ZS3, were made with reference to the actual dimensions of the structure. Details of the mortise and tenon nodes are shown in Figure 5.

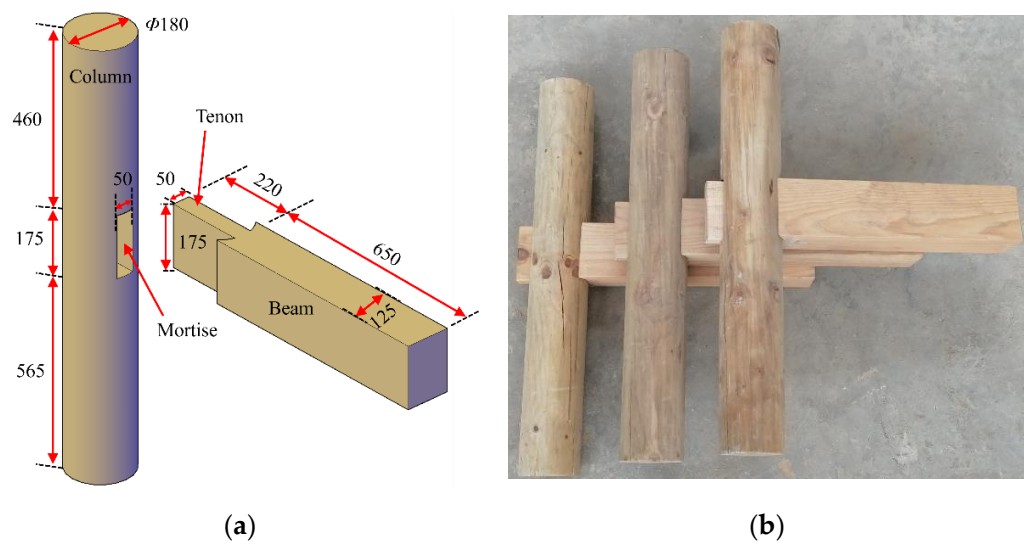

(**a**)            (**b**)

**Figure 5.** Geometric dimensions of unidirectional straight-tenon joints. (**a**) Geometric dimensions (unit: mm); (**b**) Fabricated specimens.

## 2.3. Loading and Measurement Scheme

In order to prevent the structure from overturning, both The Standard for Design of Timber Structures [31] and the Technical Standard for Maintenance and Strengthening of Historic Timber Buildings [32] stipulate that the structural inter-story displacement angle limit is 1/30 under rare earthquakes, which requires a maximum joint rotation of not over 0.033 rad. Therefore, this paper will focus on the mechanical properties of the joint within the variation range of 0–0.05 rad. The low-cycle reciprocating test is conducted using displacement-controlled loading. The loading displacement amplitude at the first stage is 5 mm, and subsequent stage increases by an additional 5 mm until reaching 30 mm (the rotation close to 0.05 rad). A total of 6 loading stages are performed, and each level undergoes three cycles of reciprocation, as illustrated in Figure 6a.

The loading point is 500 mm from the column edge on the beam. As shown in Figure 6b, a 3t force transducer is used to measure the reaction force at the loading place and a displacement gauge (W3) is used to measure the actual deformation at the loading position. Displacement gauges W1 and W2 are set horizontally at 100 mm from the top and bottom of the beam to monitor the tenon pull-out. In addition, five strain gauges, numbered S1–S5, are pasted near the tenon and mortise, and their distribution locations are shown in Figure 6c. Force transducers, displacement gauges and strain gauges were connected to the data collection instrument in full bridge, half bridge, and quarter bridge modes, respectively.

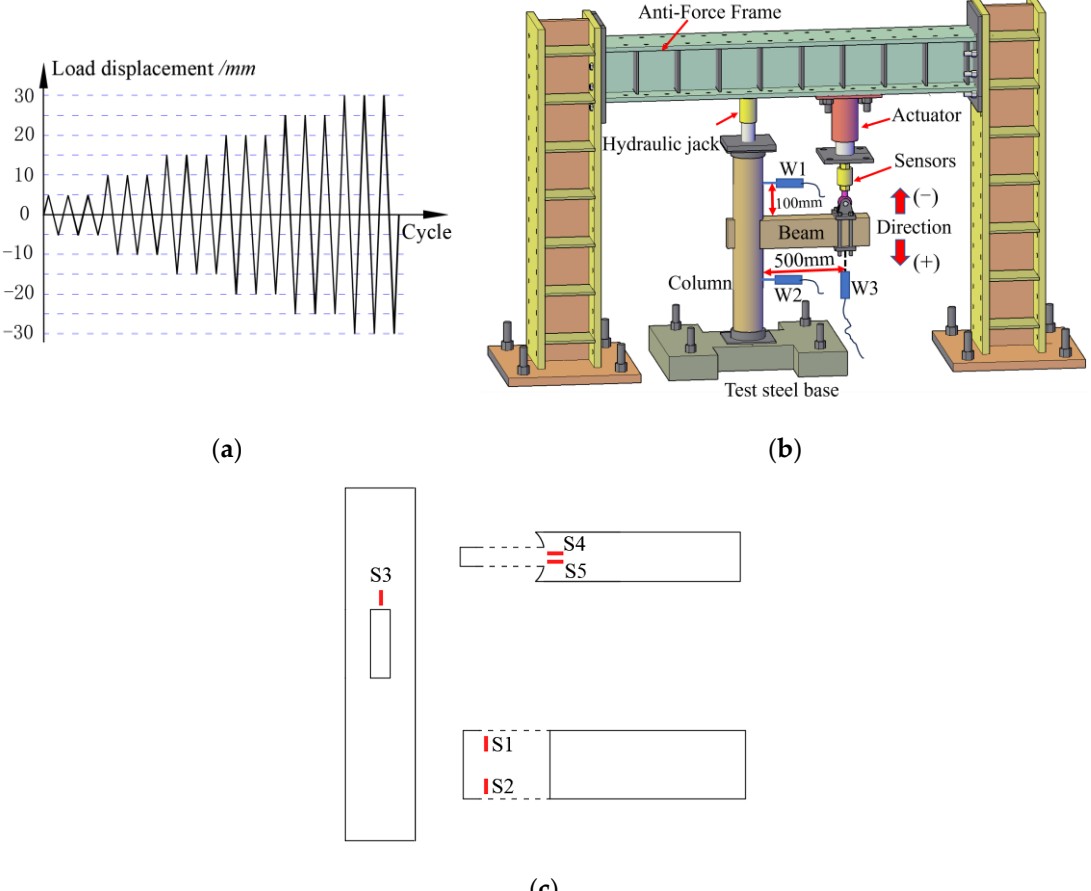

**Figure 6.** Loading setup and measurement program. (**a**) Test loading scheme; (**b**) Low-cycle recipro-cating loading tests; (**c**) Arrangement of strain gauges.

## 3. Results and Analysis

### 3.1. Test Phenomenon

The test phenomena of all T-joint specimens were similar, accompanied by a "creaking" sound during the whole test, with a louder sound when the loading increased in amplitude, and a lower sound during the unloading process. The joint deformation was small in the first two loading levels (5 mm and 10 mm), and the following pattern was significant after the third level. When the beam was loaded in the forward direction, the tenon was rotated at the point O, as shown in Figure 7a. The upper edge of the tenon slid to the right to the mortise, resulting in the tenon pulling out, with an inverted triangular distribution along the height of the beam. During reverse loading, the beam rotated at the point O, as shown in Figure 7b, and the lower edge of the tenon slid to the right, resulting in the tenon pulling out in a positive triangular distribution along the height of the beam. However, it is assumed that the deformation center of rotation of the node is the intersection of beam-column axes in the literature [21,24,25], which is inconsistent with the actual deformation characteristics of a mortise and tenon joint. With the increase of the rotation deformation, the gap between the tenon and mortise increased step by step (see Figure 8). It was also observed that wood chips fell from the gap during the test.

### 3.2. Moment-Rotation Hysteresis Curves

In this paper, the actuator's downward loading is positive, and the opposite is negative. The mortise and tenon joint's bending moment ($M$) and rotation angle ($\theta$) can be calculated from Equations (1) and (2).

$$M = F \cdot L \tag{1}$$

$$\theta = \delta / L \qquad (2)$$

where: $F$ is the reaction force measured at the loading point; $L$ is the distance from the loading point to the intersection of the beam and column axes; $\delta$ is the vertical displacement at the loading point. The moment-rotation hysteresis curve of the joint specimen is shown in Figure 9.

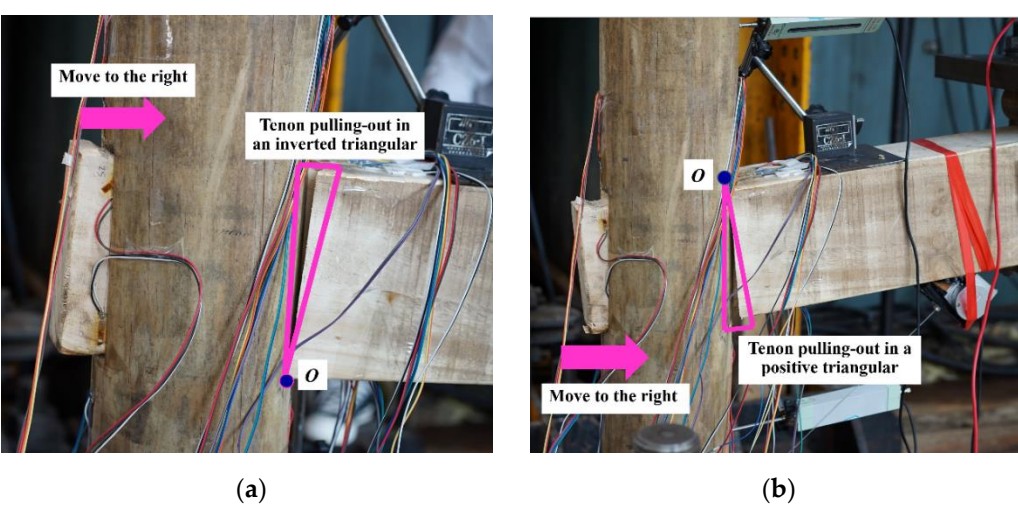

(**a**)           (**b**)

**Figure 7.** Forward and reverse loading deformation diagram. (**a**) Forward loading tests; (**b**) Reverse loading.

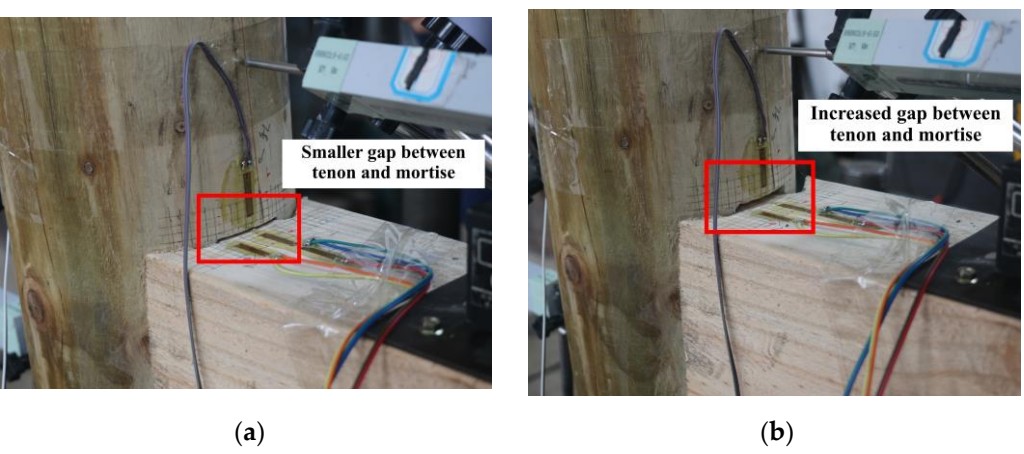

(**a**)           (**b**)

**Figure 8.** Joint gap. (**a**) Level 3 loading; (**b**) Level 6 loading.

It can be seen that the overall shape of the moment-rotation hysteresis curves of all specimens presents an inverted "Z" shape, and the "pinching" zone starts to appear in the middle of the second loading hysteresis curve. As the angle deformation increases, the "pinching" section of the hysteresis curve gradually extends and develops. This is due to the gradual accumulation of plastic deformation between the tenon and mortise, leading to an increase in the internal gap and a more extensive range of the tenon's free rotation. The area of the hysteresis curve gradually enlarges with the rotation angle increasing. The hysteresis curve area in the first cycle of the same loading level is more significant than that in the last two cycles, indicating that the plastic compressional deformation of the tenon and mortise mainly occurred during the first loading process of each level.

The moment varies with the rotation of three straight-tenon joint specimens under cyclic loading, which can be approximately simplified by several straight lines, mainly consisting of four stages: ascending (AB, BC), stress relaxation (CD), descending (DE), and

sliding (EA), as shown in Figure 10. During the first stage, the mutual extrusion and friction between tenon and mortise occurred, and the bending moment grew with the rotation. In the second stage, the joint rotation remained constant while the value of the bending moment decreased slightly. In the third stage, the bending moment decreased rapidly and tended to zero as the rotation decreased. In the last stage, the rotation gradually decreased while the bending moment value remained constant.

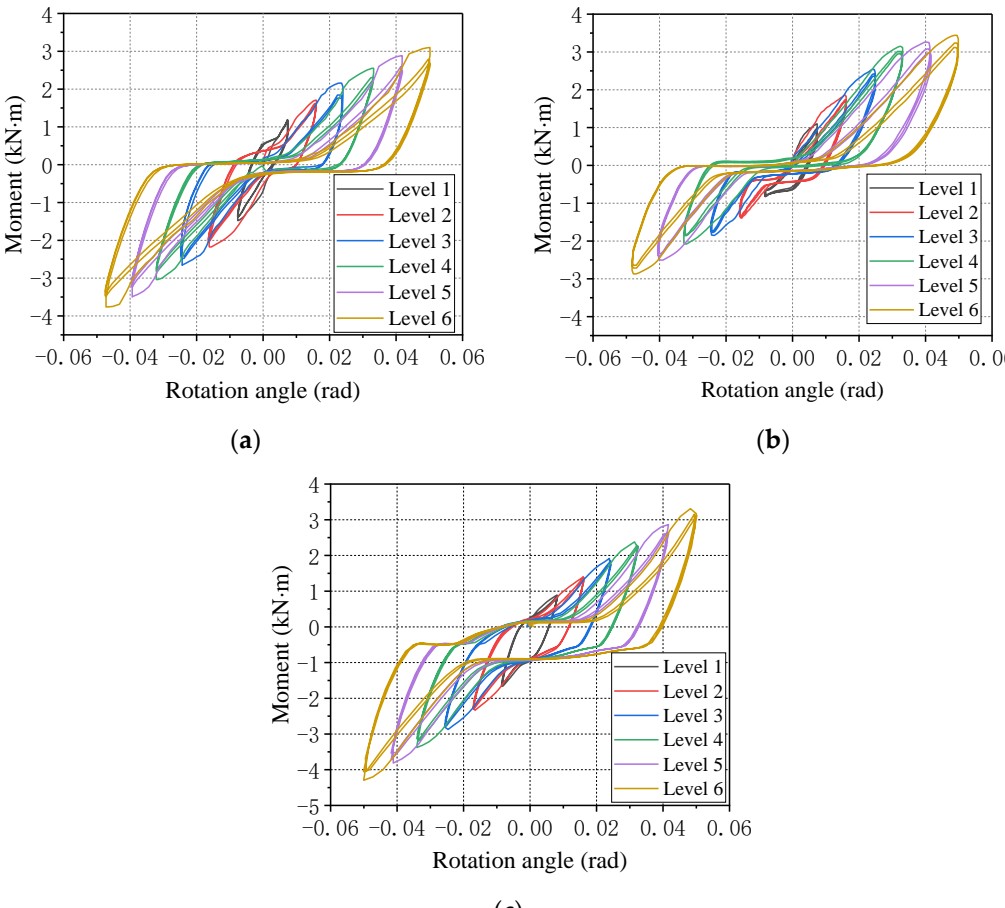

**Figure 9.** Moment-rotation hysteresis curve of the straight-tenon joint. (**a**) ZS1; (**b**) ZS2; (**c**) ZS3.

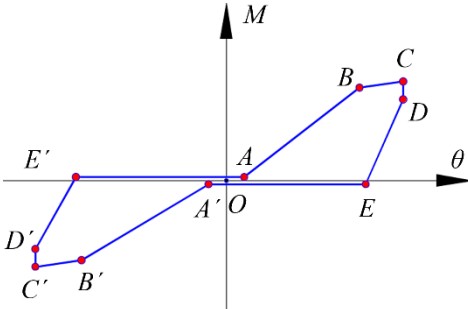

**Figure 10.** Joint hysteresis model.

### 3.3. Skeleton Curves

The skeleton curves of unidirectional straight-tenon joint specimens are shown in Figure 11. From the analysis it can be concluded that: (1) The skeleton curves have the same trend, and the shape of the whole curve is "S", so there is an apparent non-linear relation between the joint moment and rotation; (2) The bending moment values of the

specimens increased with the rotation and did not decrease under both forward and reverse cyclic loading, which indicates that the straight-tenon joint has good deformation capacity; (3) There is a specific difference in the bearing capacity of each specimen. ZS1 and ZS3 bending moment values are relatively close, and their load carrying capacity is slightly lower than that of ZS2 under forward loading, while it is significantly higher than that of ZS2 under reverse loading. The reason for this phenomenon is the different tightness of the joints caused by processing errors, and it is also closely related to the discrete characteristics of the wood.

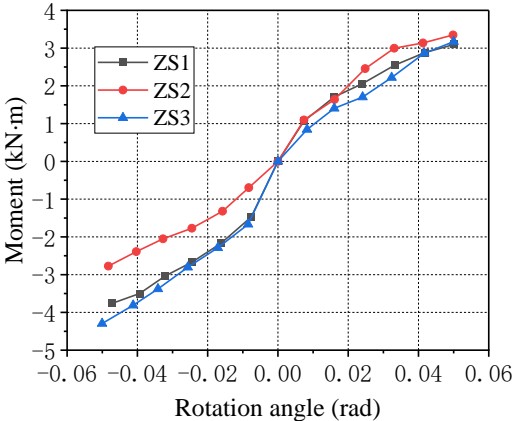

**Figure 11.** Skeleton curve of straight mortise and tenon joint.

### 3.4. The Rigidity of the Mortise–Tenon Joint

The rigidity of joint forward and reverse cut lines are calculated according to Equation (3), where taking the average value of the cut line rigidity under three cycles at each level as the rigidity of the deformation at that level, the curve of the specimen's rigidity versus rotation is obtained, as in Figure 12.

$$K_i = M_i / \theta_i \tag{3}$$

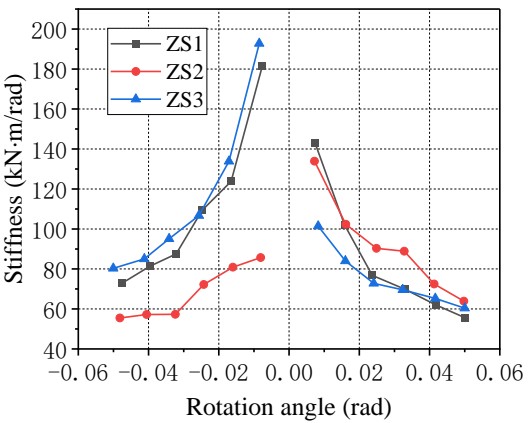

**Figure 12.** Stiffness variation curves of the straight-tenon joints.

Here, $M_i$ is the ultimate bending moment value of the $i$ cycle of the positive (negative) cycle under the $i$ level of loading, and $\theta_i$ is the rotation value corresponding to $M_i$.

From Figure 12 it can be seen that there are differences between the positive and negative rigidity of the three joint specimens. The stiffness of specimens ZS1 and ZS3 are closer, and their forward stiffness is smaller than the reverse direction, while the forward stiffness of specimen ZS2 is larger than the reverse direction. The primary change trend of the curves is the same, and the rigidity decreases gradually with the rotation increasing. The rigidity degrades relatively fast before the joint angle reaches 0.03 rad, after which the rigidity degrades relatively slowly and tends to level off. From the analysis of the

deformation and force characteristics of mortise and tenon joints, it can be seen that when the joint rotates, the mortise perpendicular to the grain and the tenon parallel to the grain are compressed. As the rotation deformation increases, the plastic compression deformation of the mortise-tenon joint increases, leading to a gradual increase in the internal gap and severe attenuation of rigidity.

### 3.5. The Rigidity of the Mortise–Tenon Joint

The equivalent viscous damping coefficient of the joint can be obtained by analyzing the joint moment-rotation hysteresis curve, calculated as follows:

$$h_e = \frac{S}{2\pi(S_{\Delta CEO} + S_{\Delta DFO})} \tag{4}$$

Here, the numerator is the area of the hysteresis curve (shaded part in Figure 13) and the bracketed part in the denominator is the area sum of the triangular CEO and DFO. The equivalent viscous damping coefficient of the joint under the rotation is calculated as the average of the results of the three hysteresis curves per stage, and the resulting curve is shown in Figure 14.

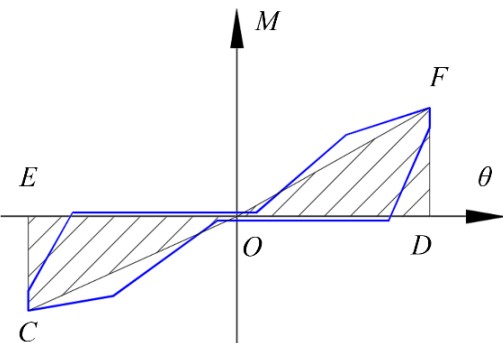

**Figure 13.** Schematic diagram of equivalent viscous damping coefficient calculation.

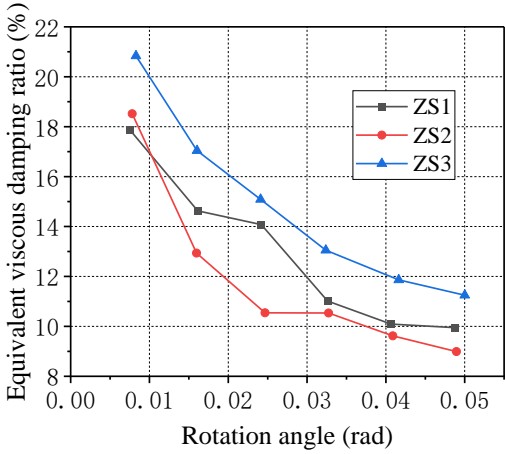

**Figure 14.** Equivalent viscous damping coefficient of the joint versus the rotation.

From Figure 14, we can see that the energy dissipation capacity of the three joint specimens is relatively close, and the equivalent viscous damping coefficient decreases with the rotation angle increase. The decrease is fast until the rotation angle reaches 0.03 rad and then slows down. This phenomenon can be analyzed from Formula (4) where the numerator term is the dissipated energy of the joint, and is determined by the friction between the tenon and mortise, as well as the plastic extrusion deformation, with the area enclosed by the hysteresis curve as its value. The energy dissipation under each loading

is shown in Figure 15. It can be seen that the joint energy dissipation gradually increases as the rotational deformation expands. The denominator term of Formula (4) is the elastic energy of the joint. As the deformation increases, the compression and friction between the tenon and mortise increases, the load carrying capacity increases, and the elastic strain energy increases. Although the numerator and denominator increase at the same time, the equivalent viscous damping coefficient ($h_e$) decreases. This result is due to the severe "pinching" effect of the hysteresis loop, which makes the increase of the energy dissipation lower than the growth rate of the elastic energy.

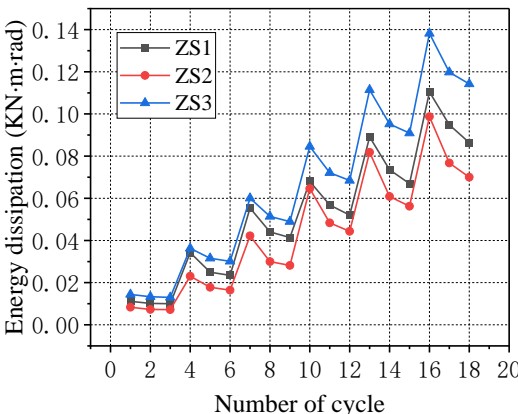

**Figure 15.** Energy dissipation of joints under each cycle.

### 3.6. Tenon Pull-out and Strain Analysis

The residual extraction value δ of the tenon can be calculated according to Equation (5):

$$\delta = \frac{\delta_1 + \delta_2}{2} \tag{5}$$

where $\delta_1$ and $\delta_2$ are the values measured by the displacement sensors W1 and W2, respectively. The relationship between the residual tenon pull-out value of the node and the corner angle is shown in Figure 16. It can be seen that the tenon node has a certain amount of tenon pull-out with respect to the initial position after repeated loading, and the residual tenon pull-out increases gradually with the increase of the corner deformation.

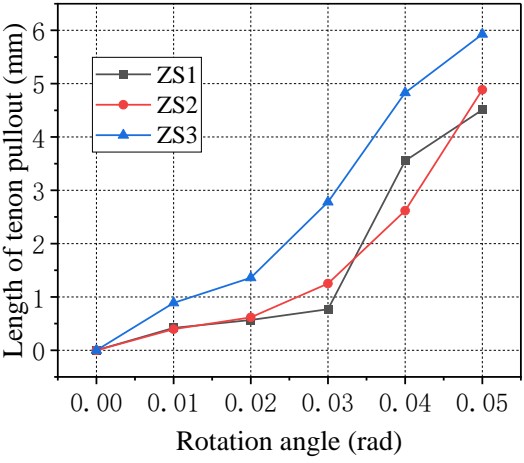

**Figure 16.** Relationship between the amount of tenon pullout and rotation.

Strain gauges were placed near the region of the mortise nodes. Some of the strain gauges stopped working during the test, and the data numbered S1 and S3 were relatively complete. The results are shown in Figure 17. During the loading process, the mortise and

tenon extruded into each other, and the compressive strains are at S1 and S3. With the increase of corner deformation, the strain value of S3 increases slowly, while the strain value of S1 grows faster and is much larger than that of S1, which indicates that the deformation of mortise is relatively small under the reciprocating loading and that the deformation of the tenon is very significant.

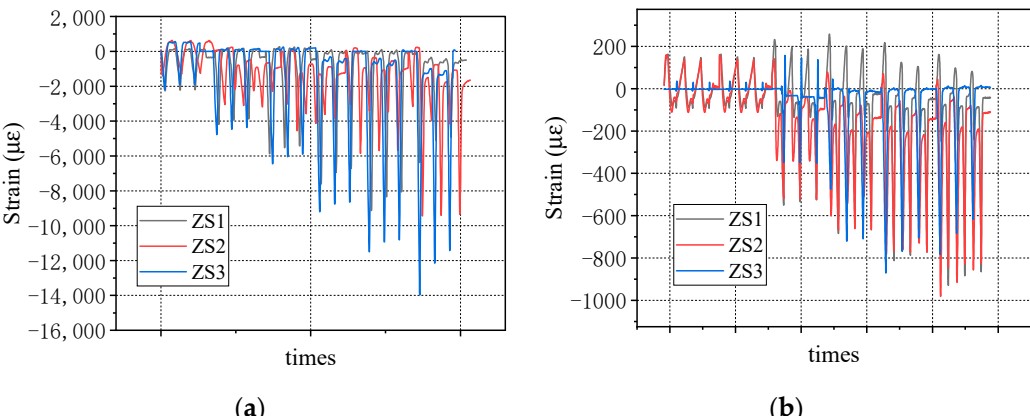

(**a**)                 (**b**)

**Figure 17.** Variation of strain values under loading. (**a**) S1; (**b**) S3.

## 4. Analytical Study

### 4.1. Basic Assumptions

The actual load states of mortise-tenon joints during rotation are extremely complex, and affected by many factors. To establish a simplified and practical joint moment rotation model, the variables less influential to the bearing capacity of mortise-tenon joints are excluded, and the following basic assumptions are made:

(1) Assume that the tenon only undergoes rigid body motion within the mortise by ignoring the tenon bending and shear deformation;

(2) Assume the friction coefficient between the tenon and the contact surface of the mortise top and bottom is a fixed constant by ignoring the friction between the tenon and the mortise side, as well as the mortise neck and the column side;

(3) When the mortise and tenon joint rotate, the tenon is in the compression perpendicular to grain, and the mortise is in the compression parallel to grain. According to the material characteristic test, the elasticity modulus of parallel-to-grain is much larger than that of perpendicular-to-grain. Therefore, assuming the direction parallel to the grain of the wood is rigid, only the tenon perpendicular to grain is deformed;

(4) As in Figure 18, with the relatively less increase in the perpendicular-to-grain compressive stress during the plastic phase, the stress-strain relation in the compression perpendicular to the grain of the wood can be assumed to be the ideal elastoplastic model.

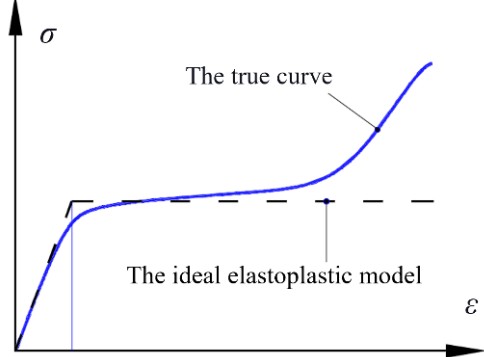

**Figure 18.** Stress-strain relation in compression perpendicular to grain of wood.

### 4.2. Derivation of Moment-Rotation Model for Straight-Tenon Joints

### 4.2.1. Model Parameters and Deformation Force Analysis

The dimensions of the unidirectional straight-tenon joint are: $h$—the height of the tenon, $b$—the width of the tenon, $d$—the diameter of the post, $g$—the gap between the tenon and the mortise, $\mu$—the friction coefficient between the contact surfaces of the wood, $E$—the modulus of elasticity compression perpendicular to grain, $\delta_y$—the ultimate deformation of elasticity in compression perpendicular to grain, and $\theta$—the joint rotation.

Since the straight-tenon joint is symmetrical in the forward and reverse directions, this paper analyzes only the clockwise rotation of the joint as an example, and the deformation process is as follows. Before the angle of rotation reaches $\theta_0$, the tenon is rotated with point $c$ in Figure 19a as a fixed center. Due to the gap, the tenon and the mortise do not interact. After the angle deformation go beyond $\theta_0$, the tenon and the mortise (assumed rigid) extrude each other and form a triangular compression zone at the top and bottom of the tenon. At the same time, the center of rotation gradually moves vertically from point $c$ to point $c'$. It can be seen that the tenon is not subjected to any force at $\theta < \theta_0$, but subjected to the combined forces $P_t$ and $P_b$ in the top and bottom pressure zones at $\theta > \theta_0$. In addition, the relative slip exists in the top pressure zone, and the tenon is also subjected to the horizontal leftward frictional force $f$, as shown in Figure 19b, where $\theta_0$ is calculated as follows:

$$\theta_0 = arc\tan(g/d) \tag{6}$$

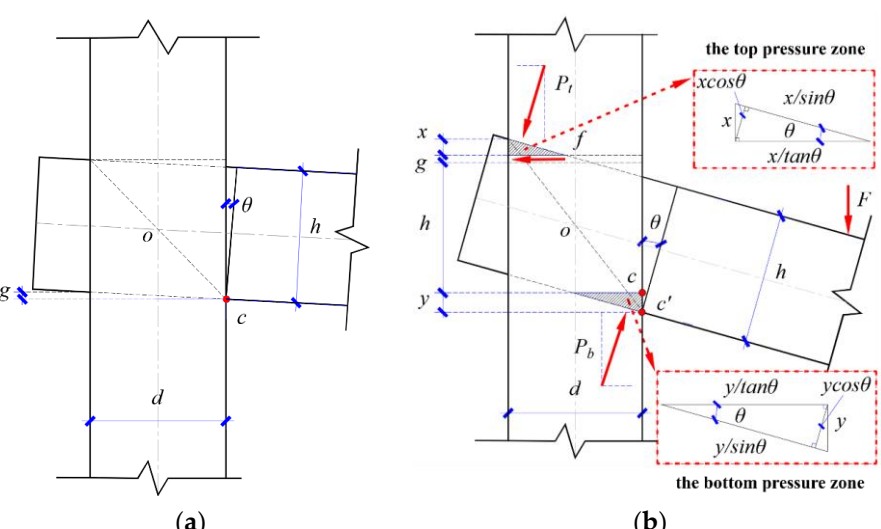

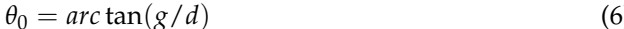

**Figure 19.** Deformation process and force analysis. (**a**) No compressional deformation; (**b**) Compressional deformation and force.

### 4.2.2. Equilibrium Conditions

According to Figure 19b, taking the tenon as the object of study, by the geometric deformation conditions and static equilibrium conditions, the following can be obtained:

$$x + y + h + g = d\tan\theta + h/\cos\theta \tag{7}$$

$$\mu P_t \cos\theta + P_t \sin\theta = P_b \sin\theta \tag{8}$$

### 4.2.3. The Equivalent Force of Triangular Compression Zone

When the tenon compressive zone is in the elastic deformation stage, the stress is triangularly distributed, the compressive stresses in the compression zone can be equated to a concentrated force $P_E$. For a simplified calculation, we consider that the point of force $P_E$ is at $1/3$ of the side of the triangle, and the direction of force is perpendicular to the grain, as shown in Figure 20a. After the compressive zone enters the elastoplastic deformation stage,

the stress beyond the deformation site of the ultimate elastic deformation will not increase since the stress-strain relation in compression perpendicular to the grain of the wood is the ideal elastoplastic model. The stress in the compression zone shows a trapezoidal distribution, which can be equated to a concentrated force $P_P$. Similarly, using the same method as for force $P_E$, we determine the location of the equivalent force $P_P$, as shown in Figure 20b. The equivalent forces of the compression zone's elastic and elastoplastic deformation stages are $P_E$ and $P_P$, respectively. The specific formulas are as follows:

$$P_E = \frac{1}{2} E \varepsilon b \Delta / \tan \theta = \frac{E b \Delta^2}{2h \tan \theta} \tag{9}$$

$$P_P = Eb\left(\frac{\Delta \delta_y}{h \sin \theta} - \frac{\delta_y^2 \cot \theta}{2h} - \frac{\delta_y^2 \tan \theta}{2h}\right)V \tag{10}$$

In order to obtain the moment carrying capacity of the mortise, the distance from the point of equivalent force action to point $O$ needs to be calculated. The distances of the equivalent forces $P_t$ and $P_b$ to point $O$ are $R_t$ and $R_b$, respectively. The distance between the friction force in the top pressure zone and the point $O$ is $R_f$, and the angle between the diagonal of the mortise and the bottom edge is $\varphi$, as shown in Figure 21. These distances can be obtained from the geometric relationship:

$$R_t = 0.5\sqrt{d^2 + (h+g)^2} \cos(\varphi - \theta) - \frac{x \cos^2 \theta}{3 \sin \theta} \tag{11}$$

$$R_b = 0.5\sqrt{d^2 + (h+g)^2} \cos(\varphi - \theta) - \frac{y \cos^2 \theta}{3 \sin \theta} \tag{12}$$

$$R_f = 0.5(g+h) \tag{13}$$

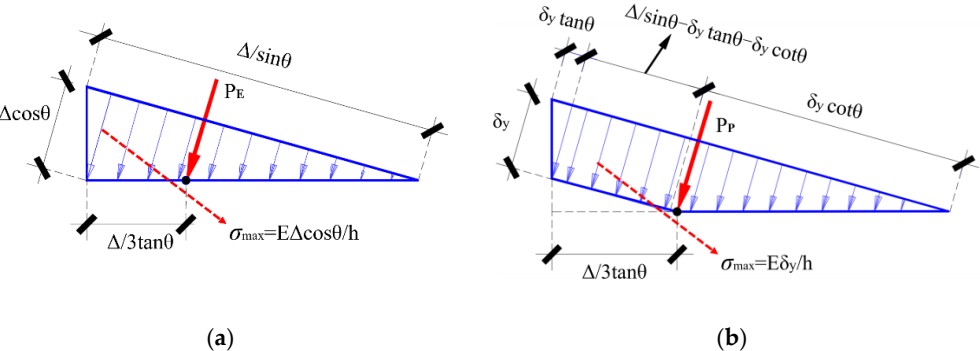

(**a**)  (**b**)

**Figure 20.** Distribution of compressive stress in the triangular compressive zone. (**a**) Elastic deformation stage; (**b**) Elastic-plastic deformation stage.

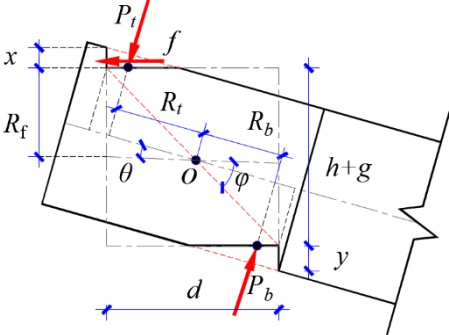

**Figure 21.** Force arm calculation diagram.

### 4.2.4. Calculation of the Resultant Force in the Compressive Zone at Different Deformation Stages

At the initial stage $0 \leq \theta \leq \theta_0$, no compressional deformation occurs between the tenons and mortises, and the joint bending moment value $M_0$ is 0.

When the joint rotation $\theta > \theta_0$, the top and bottom of the tenon are compressed. The force analysis of the beam shows that the deformation of the bottom compressive zone is slightly more significant and enters the elastic-plastic deformation stage earlier than the top compressive zone. The complete deformation process of the tenon is divided into three stages. During the first stage of $i$ ($i$ = I, II, III), the maximum vertical deformation of the imbedded compression in the top tenon is set as $x_i$, whose resultant force is $P_{i,t}$. The maximum vertical deformation of the bottom imbedded compression is set as $y_i$, whose resultant force is $P_{i,b}$. The resultant force of the top and bottom tenons in the three stages is calculated as follows:

Phase I: $\theta_0 < \theta \leq \theta_1$ ($\theta_1$ is the corresponding rotation at $y\cos \theta = \delta_y$), the top and bottom of the tenon are only deformed elastically, and the following can be obtained through these simultaneous Equations (7)–(9):

$$x_{\mathrm{I}} = \frac{(h/\cos \theta + d \tan \theta - h - g)\left(\sqrt{\sin^2 \theta + \mu \sin \theta \cos \theta} - \sin \theta\right)}{\mu \cos \theta} \tag{14}$$

$$y_{\mathrm{I}} = (h + d \sin \theta)/\cos \theta - h - g - x_{\mathrm{I}} = \frac{(\mu \cos \theta - \sqrt{\sin^2 \theta + \mu \sin \theta \cos \theta} - \sin \theta)(h/\cos \theta + d \tan \theta - h - g)}{\mu \cos \theta} \tag{15}$$

$$P_{\mathrm{I},t} = \frac{1}{2}E\varepsilon bx/\tan \theta = \frac{Ebx_{\mathrm{I}}^2}{2h \tan \theta} \tag{16}$$

$$P_{\mathrm{I},b} = \frac{1}{2}E\varepsilon bx/\tan \theta = \frac{Eby_{\mathrm{I}}^2}{2h \tan \theta} \tag{17}$$

Phase II: $\theta_1 < \theta \leq \theta_2$ ($\theta_1$ is the corresponding rotation at $y\cos \theta = \delta_y$ and $\theta_2$ is the corresponding rotation at $x\cos \theta = \delta_y$). The tenon bottom area enters elastoplastic deformation, and the top area still is in elastoplastic deformation, and the following can be obtained through these simultaneous Equations (7)–(10):

$$x_{\mathrm{II}} = \frac{-\delta_y + \sqrt{\left(\frac{2\mu \cos^2 \theta}{\sin \theta} + 2 \cos \theta\right)(d \tan \theta + h/\cos \theta - h - g)\delta_y - \mu \cot \theta \delta_y^2}}{\frac{\mu \cos^2 \theta}{\sin \theta} + \cos \theta} \tag{18}$$

$$y_{\mathrm{II}} = d \tan \theta + h/\cos \theta - h - g - x_{\mathrm{II}} \tag{19}$$

$$P_{\mathrm{II},t} = \frac{Ebx_{\mathrm{II}}^2}{2h \tan \theta} \tag{20}$$

$$P_{\mathrm{II},b} = Eb\left(\frac{y_{\mathrm{II}}\delta_y}{h \sin \theta} - \frac{\delta_y^2 \cot \theta}{2h} - \frac{\delta_y^2 \tan \theta}{2h}\right) \tag{21}$$

Phase III: $\theta > \theta_2$ ($\theta_2$ is the rotation at $x\cos \theta = \delta_y$). The top and bottom of the tenon are both in elastoplastic deformation, and the following can be obtained through these simultaneous Equations (7), (8) and (10):

$$x_{\mathrm{III}} = \frac{\frac{\mu \delta_y}{2 \sin \theta} + d \tan \theta + h/\cos \theta - h - g}{\mu \cot \theta + 2} \tag{22}$$

$$y_{\text{III}} = d \tan\theta + h/\cos\theta - h - g - x_{\text{III}} \tag{23}$$

$$P_{\text{III},t} = \frac{Eb\left(\frac{2x_{\text{III}}\delta_y}{\sin\theta} - \delta_y^2 \cot\theta - \delta_y^2 \tan\theta\right)}{2h} \tag{24}$$

$$P_{\text{III},b} = \frac{Eb\left(\frac{2y_{\text{III}}\delta_y}{\sin\theta} - \delta_y^2 \cot\theta - \delta_y^2 \tan\theta\right)}{2h} \tag{25}$$

#### 4.2.5. Theoretical Mode of Moment-Rotation (*M-θ*)

In summary, the *M-θ* relation of the unidirectional straight-tenon joint can be composed of four parts as follows:

$$M = \begin{cases} M_0 = 0, 0 \le \theta \le \theta_0 \\ M_{\text{I}} = \mu P_{\text{I},t} \cos\theta R_f + P_{\text{I},t} R_t + P_{\text{I},b} R_b, \theta_0 < \theta \le \theta_1 \\ M_{\text{II}} = \mu P_{\text{II},t} \cos\theta R_f + P_{\text{II},t} R_t + P_{\text{II},b} R_b, \theta_1 < \theta \le \theta_2 \\ M_{\text{III}} = \mu P_{\text{III},t} \cos\theta R_f + P_{\text{III},t} R_t + P_{\text{III},b} R_b, \theta > \theta_2 \end{cases} \tag{26}$$

#### *4.3. Theoretical Model and Test Comparison*

The slippage friction coefficient between woods ranges from 0.10 to 0.65 [33], and the friction coefficient $\mu$ took the median value of 0.3 in this paper. Combining the wood material characteristic test results and the specimens' geometry in the previous section, the specimens' theoretical moment-rotation curve can be obtained using the *M-θ* formula derived in the previous section. Due to factors such as specimen processing errors and discrete wood properties, there are specific differences in the experimental values of the two specimens under forward and reverse loading. Taking the average of the absolute values of the forward and reverse moments as the value for each level of deformation at the joint, the theoretical model is compared with the experimental moment-rotation curve, as shown in Figure 22.

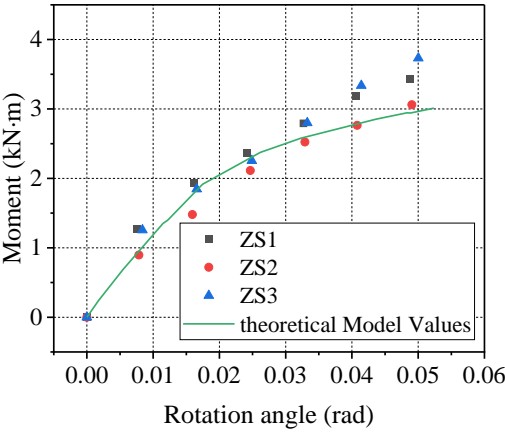

**Figure 22.** Comparison of theoretical model and test results.

From Figure 22, it can be seen that the test values of the three specimens are distributed around the theoretical curves, indicating that the theoretical model can effectively reflect the trend of straight-tenon joints' bending moment. At the same time there are some differences between theory and test, mainly closely related to the discrete properties of wood and errors in the specimen production process. In addition, the theoretical model assumes that the compressive stress-strain relationship of the wood transverse grain is ideal elastic-plastic, which cannot take into account the strengthening effect of the material after yielding, and the theoretically calculated value will be slightly lower than the test value when the rotational deformation is larger. Compared with the theoretical models

in the existing literature [18–20,22,23], the theoretical model proposed in this paper is able to consider the actual deformation characteristics of mortise and tenon joints. The model only contains parameters such as the joint geometry, the modulus of elasticity of the perpendicular-to-grain, and the compressive strength of the perpendicular-to-grain, which is more widely applicable. Moreover, the evaluation results of the theoretical model are closer to the test, which has certain value for engineering application.

## 5. Conclusions

From the tests and theoretical studies of unidirectional straight-tenon joints in traditional wooden structures, the following conclusions can be drawn:

(1)  The moment-angle hysteresis curve of the straight-tenon joint under reciprocating loading has a significant change rule, which consists of four stages: ascending, stress relaxation, descending, and sliding;

(2)  The moment-carrying capacity of straight-tenon joints grows gradually with the rotation deformation, while the compressive deformation between the tenon and the mortise accumulates and the internal gap increases, leading to a serious reduction in stiffness;

(3)  Straight-tenon joints dissipate energy through plastic compressive deformation and friction between the tenon and mortise, and the energy dissipation increases as the rotation deformation increases. At the same time, the "pinching" phenomenon of the hysteresis curve is severe, and the decrease of the equivalent viscous damping coefficient indicates that the energy dissipation capacity of the joint is gradually degraded;

(4)  Based on the geometric deformation and static equilibrium conditions, a moment-rotation ($M$-$\theta$) theoretical model is derived for each stage, and the moment-carrying capacity is mainly determined by the joint geometry and the mechanical properties of the wood. The predictions of the theoretical model are close to the tests and can provide a theoretical basis for the overall seismic analysis of traditional wood structures.

**Author Contributions:** Conceptualization, S.Y. and W.P.; FORMAL analysis, W.P.; funding acquisition, and L.Y.; investigation, S.Y.; methodology, S.Y.; project administration, W.P.; resources, H.S.; validation, H.S.; writing—original draft, S.Y.; writing—review & editing, S.Y. All authors have read and agreed to the published version of the manuscript.

**Funding:** This research was funded by the National Key Research and Development Project of China (No. 2020YFD1100703-04) and the Yunnan Provincial Education Department Scientific Research Fund Project (2022J0066).

**Data Availability Statement:** Data are contained within the article.

**Acknowledgments:** The authors would like to express their gratitude to the teachers and employees at the Earthquake Engineering Researching Center of Yunnan for their assistance and support.

**Conflicts of Interest:** The authors declare no conflict of interest.

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
