# Peer review of "Study on Reciprocating Loading Tests and Moment-Rotation Theory of Straight-Tenon Joints in Traditional Wooden Structures"

_forests, doi:10.3390/f14122424_

Round 1
Reviewer 1 Report
Comments and Suggestions for Authors
The authors conducted testing of mechanical properties of straight tenon joints in traditional wooden structures. The authors made three samples of straight T-shaped tenon joints in accordance with the actual structure and subjected them to reciprocating load tests.
1. In table 1, please add: Shear strength parallel to grain fv. Please perform cross-grain shear strength tests.
2. To obtain the compressive stress-strain relationship perpendicular to the fibers, tests were carried out in various places of the journal. According to what standard?
3. Please include a Stress-strain curve in your article.
4. What model was used to test the bending strength of wood? There is a nonlinear elastic-plastic theory of wood.
5. What method was used to determine the yield strength in the tests?
6. Please provide information about the strain gauges used in the research. How were the directions of placement (gluing) of thermometers selected?
7. What was the measuring system for strain gauges: quarter bridge, half bridge?
8. The strength of wood without defects can be measured with high accuracy and low variability, which cannot be said about elastic moduli. Please respond to this thesis.
9. The range of nonlinearity of wood is orders of magnitude larger than that resulting from the natural range of nonlinearity of stress and true strain (true stress, true strain). Please respond to this thesis.
Article on a similar topic: https://doi.org/10.3390/buildings13071839
Reviewer 2 Report
Comments and Suggestions for Authors
Manuscript ID: forests-2746184
Type: Article
Title: Study on the Reciprocating Loading Tests and Moment-Rotation Theory of Straight-Tenon Joints in Traditional Wood Structure
Authors: Shibin Yu, Wen Pan, Hexian Su, Liaoyuan Ye
Abstract
Lines 14-16: Please rephrase the sentence: “The law was analyzed by which …..”.
Lines 24-26: Please rephrase the last sentence: “The calculated results of the joint…..”
Introduction
Line 45: conducted instead of con-ducted
Lines 53-54: Instead of repeating the aim of the cited reference in the sentence “The aim is to quantitatively analyze …..” it is better to mention the authors’ conclusion.
Lines 76-78: You stated that “there are many experimental and theoretical studies on mortise-tenon joints, most of the them assume that the center of rotation….”. Please mention them and introduce them in the Reference section.
2. Materials and methods
Line 91: You don’t test the straight tenon joint specimen here, but the properties of wood to compression. Introduce data regarding the shape and the sizes of the specimens used for these tests.
Line 97 (Table 1): The measurement unit is MPa, not Mpa.
Line 114: applies instead of ap-plies
Line 137: what is “c direction”?
Line 139: observed instead of ob-served
Lines 251-252: Revise the statement: “modulus of parallel-to-grain is much larger than that of parallel-to-grain”
Line 291: PE instead of PE.
Line 298: PP instead of PP
3. Result and analysis
The results and the analysis of the results are clearly presented, but the authors didn’t comment anything about the cited references in the Introduction chapter, so to have a comparison to the results of their study.
I recommend the authors to make a connection between the cited references and the results of the study presented in the article, to justify why the cited references are important for their research.
My conclusion is that the research is interesting and deserves to be published in the journal “Forests”, but needs improvement both in the “Introduction” section and in the “Results and analysis section”. By modifying these two sections, the authors have to highlight the novelty of their research work, which not results clearly from the article.
Comments on the Quality of English Language
There are also mistakes in the writing of the work and in the ambiguous formulation of some sentences.
Round 2
Reviewer 1 Report
Comments and Suggestions for Authors
The authors took into account my comments on the article. Thank you. I have no further comments.